# Protocol for an open labelled observational study in children & adolescents with severe asthma with an eosinophilic phenotype treated with mepolizumab (CASAM)

Atul Gupta[1,2‡*], Ines de Mir Messa[3‡], Jose Valverde-Molina[4], Clare S. Murray[5], Luis Moral[6], Javier Torres Borrego[7], Katharine Pike[8], Ana Díaz-Izquierdo[3], Ana Martínez-Cañavate[9], Prasad Nagakumar[10], James Cook[1], Latika Gupta[1], On behalf of the CASAM Group[¶]

1 Paediatric Respiratory Department, King's College Hospital, Department of Women and Children's Health, London, United Kingdom, 2 Faculty of Life Sciences and Medicine, King's College London, London, United Kingdom, 3 Hospital Universitari Vall Hebron, Institut de Recerca de l'Hospital Universitari Vall d'Hebron (VHIR), Barcelona, Spain, 4 Department of Paediatrics, Santa Lucía General University Hospital, IMIB Biomedical Research Institute, Cartagena, Murcia, Spain, 5 Division of Immunology, Immunity to Infection and Respiratory Medicine, School of Biological Sciences, The University of Manchester, Manchester Academic Health Science Centre, and Manchester University NHS Foundation Trust, Manchester, United Kingdom, 6 Paediatric Allergy and Respiratory Unit, Alicante University General Hospital, Alicante Institute for Health, and Biomedical Research (ISABIAL - FISABIO Foundation), Alicante, Spain, 7 Paediatric Allergy and Respiratory Unit, Children's University Hospital Reina Sofia, and The Maimonides Institute for Biomedical Research of Cordoba (IMIBIC), Córdoba, Spain, 8 Bristol Royal Hospital for Children, Bristol, United Kingdom, 9 H.U. Virgen de las Nieves, Granada, Spain, 10 Birmingham Women's and Children's Hospital, Birmingham, United Kingdom

¶ CASAM Group: Membership of the CASAM Group is provided in the acknowledgments.
‡ These authors are joint senior authors on this work.
* Atul.gupta@kcl.ac.uk

## Abstract

### Introduction

Children with severe eosinophilic asthma are at an increased risk of severe exacerbations, medication side effects, impaired lung function, and significantly reduced quality of life. The advent of biologic agents aimed at eosinophilic pathways has significantly improved management options for patients with severe asthma. Mepolizumab, an anti-interleukin-5 monoclonal antibody, was approved in 2018 as an add-on maintenance treatment for patients with severe eosinophilic asthma aged ≥6 years of age. Mepolizumab's efficacy and safety is well documented in adults, but paediatric data remain limited. This real-world observational study will evaluate the long-term safety and efficacy of mepolizumab in children and adolescents, including those with comorbidities and on concomitant medications to provide a clearer picture of its performance in everyday clinical practice.

**Data availability statement:** No datasets were generated or analysed during the current study. All relevant data from this study will be made available upon study completion.

**Funding:** This research was supported by an Investigator-Sponsored Studies (ISS) GSK grant, with no additional contributions to the project. The funders had no role in study design, data collection and analysis, decision to publish, or preparation of the manuscript.

**Competing interests:** The authors have declared that no competing interests exist.

**Abbreviations:** ACQ: Asthma Control Questionnaire; ACT: Asthma Control Test; ADR: Adverse Drug Reaction; AE: Adverse Event; ATS: American Thoracic Society; BMI: Body Mass Index; BTS: British Thoracic Society; c-ACT: Childhood Asthma Control Test; CASI: Composite Asthma Severity Index; eCRF: electronic Case Report Form; ED: Emergency Department; EDC: Electronic Data Capture; FeNO: Fractional nitric oxide; FEV1: Forced expiratory volume in 1 second; FVC: forced vital capacity; GCP: Good Clinical Practice; GINA: Global Initiative for Asthma; GLI: Global Lung Initiative; GPP: Good Pharmacoepidemiology Practices; HCPs: Health Care Providers; HRQL: Health Related Quality of Life; ICF: Informed Consent Form; ICH: International Conference on Harmonisation; ICS: Inhaled corticosteroids; IEC: Independent Ethics Committee; IgE: Immunoglobulin E; IRB: Institutional Review Board; OCS: Oral Corticosteroids; PAQLQ: Paediatric Asthma Quality of Life Questionnaire; PK: Pharmacokinetic; SA: Severe Asthma; SAE: Serious Adverse Event; VAS: Visual Analogue Scale; WPAI: Work Productivity and Activity Impairment Questionnaire.

## Methods and analysis

Design: CASAM is an open label, multinational (Spain and United Kingdom), multi-center (15 Spanish and 6 UK public institutions) observational cohort study. Sample size: 150 paediatric patients on mepolizumab for SA between 6–17 years of age will be enrolled. Study Duration: Study will be conducted over 36 months (12 months pre mepolizumab initiation and 24 months after initiation). Data Collection: Pseudonymized data (prospective and retrospective) from medical charts will be collected and entered in the electronic case report forms (eCRFs) of the electronic data capture (EDC) system. Primary Outcome: To compare the rates of clinically significant asthma exacerbations in the pre-exposure and the 12-month post-exposure period with mepolizumab treatment. Protocol Version: CASAM Study_Protocol_V2.0_16th November 2023 KCH

## Registration details:

The study is registered with ClinicalTrials.gov (NCT05139381), IRAS number 306475.

## 1. Introduction

Around 11% of children aged 6–12 years old worldwide suffer from asthma [1–3], making it the most common chronic illness in paediatrics [4]. Although the majority of children & young people (CYP) with asthma are able to control their symptoms with available standard of care treatment, about 2–5% have severe asthma (SA) and need regular high-dose inhaled corticosteroids (ICS), a second controller, and/or chronic systemic corticosteroids for management [5–7]. A study conducted across 30 paediatric pulmonology and allergy units in Spain reported a SA prevalence of approximately 8.8% to 24.2%. Children with SA face higher risks of severe exacerbations, medication side effects, impaired lung function, worse quality of life (QoL), and increased adult chronic obstructive pulmonary disease (COPD) risk compared with mild/moderate asthma [6–14]. Adolescents with SA have more frequent hospitalizations as compared to the adults with SA [15], significant school absenteeism [16] thus a higher disease burden

Inflammation in severe asthma can be categorised as either eosinophilic or neutrophilic [5]. The eosinophilic phenotype is characterised by airway infiltration with eosinophilic inflammatory cells leading to poor asthma control, frequent exacerbations, greater risk of near-fatal asthma attacks and reduced lung function. This poses a huge burden on caregivers, increased healthcare utilisation and cost, reduce quality of life (QoL) [9,13,17]. Therefore, there remains a need for safe and effective treatment options that target diverse inflammatory pathways. Deeper understanding of asthma endotypes and phenotypes led to development of monoclonal antibodies (mAb) (Biologics) that target specific inflammatory pathways.

Mepolizumab got approved as an add on therapy for the treatment of eosinophilic SA in children ≥6 year of age in Europe in 2018. It is a humanised, anti-IL5 antibody (immunoglobulin G (IgG) Kappa) that binds with high affinity to human IL-5, thus regulates activation of the IL-5 receptor present on eosinophils and basophils [1,18]. In Phase 3 adult trials, adding subcutaneous (SC) mepolizumab to standard care significantly reduced exacerbations, decreased maintenance corticosteroid dependency, improved symptom control and health related QoL [19–25].

## Mepolizumab in 12–17 years

In Phase II and III clinical trials, due to the low prevalence of severe eosinophilic asthma in adolescents, recruitment was challenging. A total of 37 adolescent patients were randomized across the 4 mepolizumab trials [26]: the 32-week MENSA study [22] (n = 25; NCT01691521); 24-week MUSCA study (n = 9; NCT02281318) [20]; 52-week DREAM study (n = 1; NCT01000506) [27]; and 24-week SIRIUS study (n = 2; NCT01691508) [19]. Of the 37 recruited patients from MENSA and SIRIUS study, 26 entered COSMOS (NCT01842607) [21], a 52-week, open-label extension study to understand long term safety of mepolizumab. In this study all patients received 100 mg SC mepolizumab. MUPPITS-2 trial is the largest randomised trial in paediatrics which enrolled 290 children and adolescents aged 6–17 years [28]. In the adolescents, the asthma control trended in favour of mepolizumab with overall significant reduction in exacerbations. The adverse event (AE) reported (headache, nasopharyngitis, and upper abdominal pain) were broadly consistent with the overall population [19–21,23–29] and like adult studies, reduction was noted in blood eosinophil counts [19,20,22,26,27,30].

## Mepolizumab in 6–11 years

*Gupta et al*, in their multi-national, non-randomised, open-label, pharmacokinetic trial reported first clinical evaluation of mepolizumab SC in children aged 6–11 years with severe eosinophilic phenotype [31]. Mepolizumab SC half-life of 22–24 days in children was congruous with adult value of 16–22 days. Mepolizumab was deemed to be safe in children with no specific AE patterns and no new safety concerns beyond those seen previously in adults and adolescents over 52 weeks [19–21,23–25,27,31,32]. They observed sustained reduction in blood eosinophils (to 50 cells/μL) during the 12-week dosing phase in Part A [31], and during the long-term, 52-week dosing phase in Part B [32]. Blood eosinophil reductions correlated with improvements in asthma control, as shown by ACQ-7 and c-ACT scores. *Jackson et al* in their randomised controlled trial demonstrated a 30% reduction of the AER in the mepolizumab group compared to controls (AER ratio 0.73; 0.56–0.96; P = 0.027), which is less than adult studies [28]. Mepolizumab in trials hasn't demonstrated significant change in function [31,32]. Long term safety data in adults are not associated with opportunistic infections or neoplasms [23]. There is no long term (>52 weeks) safety data or real-world evidence of use of mepolizumab in children & adolescents with SA.

## 2. Rationale

SA in children and adolescents remains a challenging condition to manage. While mepolizumab, has demonstrated efficacy and safety in adult population, paediatric data are limited. Most existing paediatric studies have focused on pharmacokinetics and safety, with few addressing clinical outcomes, and many lacking a comparator arm [28,31]. Recruitment challenges and ethical concerns have also limited the feasibility of trials in this age group. The clinical effectiveness of mepolizumab in children and adolescents, particularly in routine care settings, remains underexplored.

This 'real-world' observational study in children and adolescents aims to bridge this gap and assess mepolizumab's effectiveness and safety in a more diverse paediatric population, with various co-morbidities and concomitant medications. The study will help us understand how the medication performs in real-world clinical practice compared with clinical trials.

## 3. Research question and objectives(s)

To assess the long-term benefit of mepolizumab based on the reduction of clinically significant exacerbations and side effects after long term use.

### 3.1 Primary objective

To compare the rates of clinically significant asthma exacerbations (defined as systemic OCS use and/or ED visit and/or hospitalisation) in the pre-exposure and the 12-months post-exposure period with mepolizumab treatment.

### 3.2 Secondary objective

1.  Compare the rate of mild and moderate clinically significant exacerbations in the pre-exposure and the 12-month and 24-month post-exposure periods with mepolizumab.

2.  Compare the rate of severe asthma exacerbations requiring an ED visit in the pre-exposure and the 12-month and 24-month post-exposure periods with mepolizumab.

3.  Compare the rate of severe asthma exacerbations requiring hospitalization in the pre-exposure and the 12-month and 24-month post-exposure periods with mepolizumab.

4.  Describe changes from index date (date of mepolizumab start) in OCS dose during the post-exposure period among participants on maintenance OCS.

5.  Describe asthma related healthcare resource utilization including count of hospital admissions, ED visits, visits to HCPs, and prescription medication utilisation for asthma.

6.  Compare prescription medication utilisation (pick up) of ICS & rescue medications in the pre-exposure and the 12-month post-exposure period with mepolizumab treatment.

7.  Compare the level of asthma control at baseline and the 12- and 24-month post-exposure periods.

8.  Determine and compare the level of asthma severity measured by composite asthma severity index (CASI) score at baseline and the 12- and 24-month post-exposure periods.

9.  Determine patterns of mepolizumab usage including adherence/persistence, duration of therapy, discontinuation, and reasons for discontinuation.

10. Determine incidence of Mepolizumab-related adverse events – parents/ patients/ physicians reported.

11. Describe the sociodemographic, socioeconomic, and clinical characteristics (including comorbidities) of the patients under mepolizumab treatment.

12. Describe the Health-Related Quality of Life (HRQL) of patients treated with mepolizumab at the pre-exposure and post-exposure periods.

### 3.3 Other objectives

Where data is available, additional objectives will be addressed:

1. Describe patient & parental satisfaction with mepolizumab treatment.

2. Describe health care professional satisfaction with mepolizumab treatment in general and per type of treatment (home vs hospital).

3. Describe lung function using the change in $FEV_1$ and FVC (measured in L and using GLI reference values) during the observational period (12 months pre and 24 months post mepolizumab initiation).

4. Describe the change in peripheral blood eosinophils during the post mepolizumab treatment period.

5. Describe the change in FeNO during the post treatment period.

6. Characterise and compare patterns of school absenteeism and parental work status, (productivity/absenteeism) at baseline and the 12- and 24-month post-exposure periods.

7. To assess the clinical profile of the subgroup of paediatric patients who start treatment with mepolizumab, after omalizumab discontinuation.

8. To assess changes in growth profile of the patients under treatment with mepolizumab at baseline and the 12 and 24-month post-exposure periods.

## 4. Research methods

• Design: CASAM is an open label, multinational (Spain and United Kingdom), multicenter (15 Spanish and 6 UK public institutions) observational cohort study.

• A SPIRIT schedule and overview of study design and data collection points can be found in Figs 1 and 2.

• Sample size: 150 paediatric patients on mepolizumab for SA between 6–17 years of age will be enrolled.

• Study Duration: Study will be conducted over 36 months (12 months pre mepolizumab initiation and 24 months after initiation). Data collection will take place until March 2026. Results will be expected in late 2025–2026.

### 4.1 Inclusion criteria

A patient will be eligible if all the following criteria apply:

• Parents/guardians and/or children & adolescents aged 6–17 years at mepolizumab initiation who provides his/her written informed consent to participate prior to commencing any study related activities.

• Patient with a current clinical diagnosis of severe asthma (as per GINA/ ATS/ ERS/ BTS criteria) and eosinophilic phenotype (as per mepolizumab licence)

• Patient currently under treatment or who initiates treatment with mepolizumab at the inclusion visit based on national asthma guidelines.

• Patient with relevant paper or electronic-based medical records available for 12-months prior to enrolment date/index date (date of first mepolizumab injection).

### 4.2 Exclusion criteria

A participant will not be eligible if any of the following criteria apply:

• Patient who does not meet the inclusion criteria.

• Patient who has participated in an asthma monoclonal antibodies drug interventional clinical trial in the previous 12-months to mepolizumab initiation.

### 4.3 Sample size calculation

To detect a 50% decrease in exacerbation rate with 90% power at a 5% significance level during a 12-month mepolizumab treatment study for children (6–11 years old) with severe uncontrolled eosinophilic asthma,

| Study Period | Enrolment | Post enrolment | | | | | |
|---|---|---|---|---|---|---|---|
| Activity | Baseline visit | All FU Visits | 6 m Visit | 9 m Visit | 12 m Visit | 18 m Visit | 24 m Visit |
| Inclusion/exclusion criteria | X | | | | | | |
| Informed consent | X | | | | | | |
| Assessments | | | | | | | |
| Demographics/socioeconomic characteristics | | | | | | | |
| • Age at mepolizumab initiation | X | | | | | | |
| • Gender | X | | | | | | |
| • Socioeconomic situation | X | | | | | | |
| • Weight / Height / BMI | X | X | | | | | |
| Clinical characteristics | | | | | | | |
| • Age on asthma onset | X | | | | | | |
| • Previous asthma medication | X | | | | | | |
| • ACT or c-ACT (asthma control) | X | X | | | | | |
| • CASI (asthma severity) | X | | | | X | | X |
| Asthma exacerbations | X | X | | | | | |
| Healthcare utilization<br>• Number of ED visits<br>• Number of Unscheduled Outpatient visits<br>• Number of Hospital admissions | X | X | | | | | |
| Details of current asthma status:<br>• Lung function<br>• Eosinophils<br>• FeNO | X | X | | | | | |
| Co-morbidities | X | | | | | | X |
| Tobacco smoke exposure | X | X | | | | | |
| Mepolizumab treatment | X | X | | | | | |
| Allergic condition (IgE, skin prick test or Specific IgE to aeroallergens) | X | | | | | | |
| Concurrent medications<br>• OCS<br>• ICS<br>• Rescue medication<br>• Other asthma medication | X | X | | | | | |
| Satisfaction with treatment (VAS) | X (if patient already under treatment) | | X | | X | X | X |
| School/work absenteeism | X | X | | | | | |
| Safety assessments | X | X | | | | | |
| PAQLQ | X | | | | X | | X |

**Fig 1. Illustrates a SPIRIT Schedule of the study.**

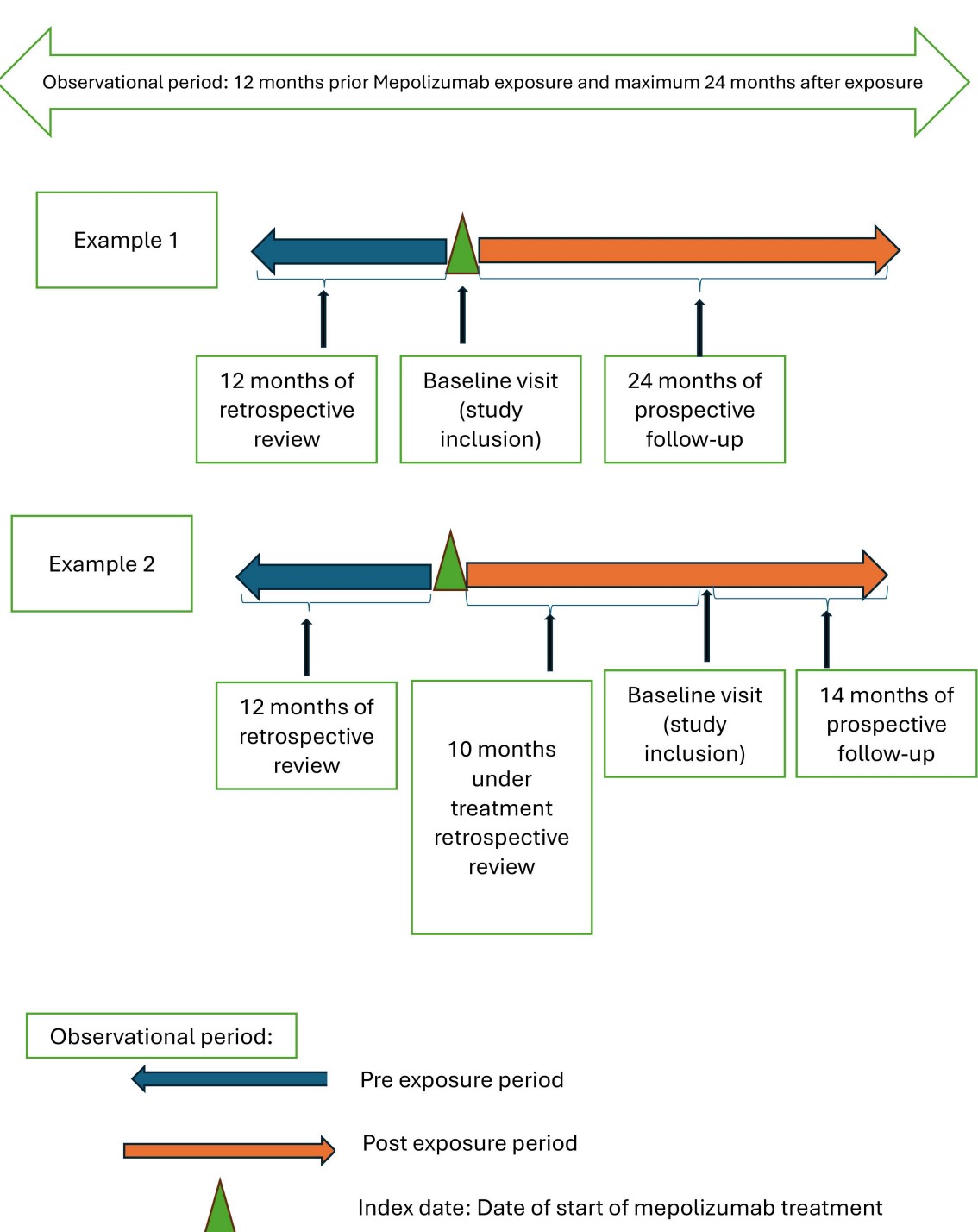

**Fig 2. Illustrates the study design and data collection points.**

assuming a 25% participant withdrawal rate (20% in the first year and 5% in the second year) and accounting for their data contribution (4 months in the first year and 18 months in the second year), you would need around 80 participants. This is based on data from study NCT02377427 (Part B), where 30 similar patients experienced a 69% (from 3.47/year to 1.09/year) reduction in exacerbation rate while receiving mepolizumab every 4 weeks for 12 months.

### 4.4  Data collection

Pseudonymized data will be recorded by nominated member(s) of the research team, into the eCRFs of the EDC system provided by the sponsor. Data in the EDC system are kept in a central location and all data will be transmitted to a central database. Confidentiality will be taken care of at each step. Each participant will have an eCRF and a unique identification number. Tables 1–3 illustrates details of data and patient reported variables collected and data collection process respectively.

   No additional study visits will be arranged, data will be collected at usual asthma healthcare visits (routine or unscheduled, remote or face to face).

   **Pre-exposure period** covers the 12-month period prior to mepolizumab initiation and is retrospectively collected.

   **Post-exposure period** covers up to 24 months after mepolizumab initiation and can be partly retrospective and partly prospective.

**Baseline data**

### 4.5  Outcome evaluation

•    Primary outcome: Clinically Significant Asthma exacerbations

The primary outcome is to calculate the rate of clinically significant exacerbations in the pre-exposure and the 12-month post-exposure period with mepolizumab treatment which is defined as deterioration in asthma requiring (1) use of systemic corticosteroids and/or (2) ED visit and/or hospital admission. Use of systemic corticosteroids is defined as OCS (e.g., prednisone) for at least 3 days or a single systemic (intravenous or intramuscular corticosteroid dose). For participants on maintenance systemic corticosteroids, at least double the existing maintenance dose for at least 3 days is required.

•    Secondary outcomes

1. Rate of clinically significant exacerbations (mild, moderate and severe) (24-month post-exposure): Evaluating the longer-term impact of mepolizumab treatment

2. ED Visits (12-month and 24-month post-exposure): Measuring the frequency of severe asthma exacerbations requiring emergency department visits

3. Hospitalizations (12-month and 24-month post-exposure): Assessing the rate of severe asthma exacerbations leading to hospital admissions

4. OCS Dose Changes: Investigating changes in OCS dosages from baseline among participants on maintenance OCS

5. Healthcare Utilization: Tracking the number of hospital admissions, ED visits, and visits to HCPs both pre-exposure and post-exposure periods

**Table 1. Illustrates baseline data collected.**

| |
|---|
| **1. Demographic characteristics:** |
| ◦ Age at mepolizumab initiation |
| ◦ Gender (male/female) |
| ◦ Weight (also collected at follow-up visits) |
| ◦ Height (also collected at follow-up visits) |
| ◦ BMI (auto calculated variable) (also calculated at follow-up visits) |
| ◦ Socioeconomic situation (number of people aged 10 or older and number of people aged 1-9/number of rooms (bedrooms and living room) at baseline |
| **2. Clinical characteristics:** |
| ◦ History of asthma: |
| ▪ Age at asthma onset |
| ▪ Degree of asthma control (c-ACT score at baseline and all follow-up visits) |
| ▪ Degree of asthma severity (CASI score at baseline, 12- and 24 months) |
| ▪ Previous medication for asthma during the 12 months before mepolizumab initiation: type, dose, discontinuation date and reason for discontinuation |
| ◦ Details of current asthma status at baseline and follow-up visits |
| ▪ Lung function data: FEV1 and FVC (if available) |
| ▪ Peripheral blood eosinophils (if available) |
| ▪ FeNO (if available) |
| ◦ History of asthma exacerbations: |
| ▪ Count exacerbations in the year before starting mepolizumab and between follow-up visits. |
| ◦ Co-morbidities at baseline (retrospectively) and at 24 months: atopic dermatitis, allergic rhinitis, obesity, food allergies, GERD, psychological factors, breathing dysfunction (EILO, hyperventilation). |
| ◦ Allergic condition (IgE, skin prick test or Specific IgE to aeroallergens) at baseline. |
| 3. Tobacco smoke exposure at baseline and follow-up visits: Yes (active/ passive)/No. |
| 4. Treatment with mepolizumab at baseline and follow-up visits: type (home/hospital), dose, dose changes, start and end date, discontinuation date and reason for discontinuation. |
| 5. Healthcare utilisation at baseline (during the 12 months before study inclusion) and between follow-up visits. |
| i. Number of Emergency Department (ED) visits |
| ii. Number of Unscheduled Outpatient visits |
| iii. Number of Hospital admissions |
| 6. Satisfaction with treatment with mepolizumab (patients/parents/HCP): VAS 0-5 (very unsatisfied to very satisfied. |
| 7. Adverse Events related to mepolizumab reported by parents/patients/physicians |
| 8. Adapted version of WPAI questionnaire for work productivity and absenteeism (to be completed by parents/legal guardian) |
| 9. Days of school absenteeism (direct question to the parents/tutors) |

6. Medication Utilization: Analysing prescription medication utilization of ICS and rescue medications during the 12-month post-exposure period with mepolizumab treatment

7. Asthma Control and Severity: Assessing asthma control and severity using the c-ACT and CASI scores at baseline, 12-month, and 24-month post-exposure

8. Mepolizumab Usage Patterns: Investigating adherence, persistence, duration of therapy, discontinuation, and reasons for discontinuation, both patient and healthcare provider perspectives

9. Adverse Events: Monitoring mepolizumab-related AE reported by parents, patients, and physicians during treatment

**Table 2. Illustrates overview of patient reported variables collected.**

| |
|---|
| 1. **c-ACT (Children Asthma Control Test)** [33]: It will be completed at baseline and follow-up visits. |
| 2. **Composite asthma severity index (CASI)** [34]: It will be completed at baseline and follow-up visits. |
| 3. **Visual Analog Scale (0–5) to measure satisfaction** with treatment for patients, parents, and HCP at baseline (in patients already under treatment) and follow-up visits at 6, 12, 18 and 24 months. |
| 4. A set of **ad-hoc questions** to collect school absenteeism: absenteeism, late arrivals, or early departures from school in the previous, missed school-related activities, rescue inhaler utilization at school in the previous 12 months before inclusion and every 3 months in the follow-up period. |
| 5. **Work Productivity and Activity Impairment Questionnaire General** (WPAI-GH) [35]: to be completed during the visits by the parents/legal guardians. |
| 6. **Paediatric Asthma Quality of Life Questionnaire** (PAQLQ): It is for children between the ages of 7 and 17 years to be completed during the visits. |

10. Patient Characteristics: Analysing sociodemographic, socioeconomic, and clinical characteristics, including comorbidities, of the study's patients

11. Health-Related Quality of Life: Measuring the patients' quality of life using the PAQLQ

### 4.6 Statistics

Descriptive statistical analysis will provide an overview of patient demographics and baseline characteristics. Continuous variables will be described using mean, standard deviation, median, percentiles, and range. Categorical variables will be summarized with frequencies and percentages. Significance will be determined using a p-value threshold of 0.05.

Primary analysis: Compare the 12-month post-exposure period with the pre-exposure period for each participant

Secondary analyses: Include both the 12-month and 24-month post-exposure periods

The exacerbation rate will be analysed using a Negative Binomial regression via generalised estimating equations with covariates of time (pre-, post-exposure) and country

All statistical analyses will be carried out using the statistical package SAS

### 4.7 Data Management

- Data entry: All data will be collected and entered directly into the electronic data capture (EDC) system

- Coding: The study will follow the current Standard Coding Instructions for coding of medical history, concomitant illness (MedDRA), concomitant medication (WHO-Drug) and adverse events/reactions (MedDRA)

- Quality: Remote monitoring by CRO of the participating sites to assess compliance, adherence, accuracy, completeness, and maintenance

- Storage: All study records, including ICFs, CRFs, SAE forms, source documents, and correspondence, are retained for at least 25 years

- Secure archiving procedures are implemented

- Analysis: Interim 12-month data analysis is foreseen

### 4.8 Ethics and dissemination

The study was initiated after approval IEC (independent Ethic Committees)/IRBs (Institutional Review Boards) from both UK and Spain and all regulations of the participating countries. The study adheres to ethical guidelines, including the

**Table 3. Illustrates data collection overview.**

| Activity | Baseline visit Retrospective data (12 months prior mepolizumab initiation and since mepolizumab initiation to inclusion visit in patients already under treatment) and current data | Follow-up visits (according to usual clinical practice, up to 24 visits are expected) *If not available in all FU visits, minimum collected at the selected visits | | | | | |
|---|---|---|---|---|---|---|---|
| | | All FU visits | 6 m | 9 m | 12 m | 18 m | 24 m |
| **Informed consent** | X | | | | | | |
| **Inclusion/exclusion criteria** | X | | | | | | |
| **Demographics/socioeconomic characteristics** | | | | | | | |
| • Age at mepolizumab initiation<br>• Gender (male/female)<br>• Socioeconomic situation | X | | | | | | |
| • Weight<br>• Height<br>• Body Mass Index (auto calculated) | X | X | | | X* | | X* |
| | **Baseline visit** Retrospective data (12 months prior inclusion) and current data | **All FU visits** | 6 m | 9 m | 12 m | 18 m | 24 m |
| **Clinical characteristics** | | | | | | | |
| History of asthma | | | | | | | |
| • Age on asthma onset<br>• Previous medication for asthma (up to 12 months prior inclusion) | X | | | | | | |
| • Degree of asthma control (c-ACT) | X | X | | | | | |
| • Degree of asthma severity (CASI) | X | | | | X | | X |
| Asthma exacerbations | X | X | | | | | |
| Healthcare utilization:<br>• Number of ED visits<br>• Number of Unscheduled Outpatient visits<br>• Number of Hospital admissions | X | X | | | | | |
| Details of current asthma status:<br>• Lung function data (if available)<br>• Peripheral blood eosinophils (if available)<br>• FeNO: fractional nitric oxide concentration (if available) | X | X | X | | X | X | X |
| Co-morbidities | X | | | | | | X |
| Active/ passive tobacco smoke exposure | X | X | | | X | | X |
| Treatment with mepolizumab | X | X | | | | | |
| Allergic condition (IgE, skin prick test or Specific IgE to aeroallergens) | X | | | | | | |
| Concurrent medications<br>• OCS<br>• ICS<br>• Rescue medication<br>• Other asthma medication | X | X | | | | | |
| Satisfaction with treatment (VAS) | X (if patient already under treatment) | | X | | X | X | X |
| School and work absenteeism | X | X | | | | | |

*(Continued)*

**Table 3.** (Continued)

| Activity | Baseline visit<br>Retrospective data (12 months prior mepolizumab initiation and since mepolizumab initiation to inclusion visit in patients already under treatment) and current data | Follow-up visits (according to usual clinical practice,<br>up to 24 visits are expected)<br>*If not available in all FU visits, minimum collected at the selected visits | | | | | |
|---|---|---|---|---|---|---|---|
| | | All FU visits | 6 m | 9 m | 12 m | 18 m | 24 m |
| Safety assessments | X | X | | | | | |
| PAQLQ (Paediatric Asthma Quality of Life Questionnaire) | X | | | | X | | X |

Declaration of Helsinki, GPP, ISPE GPP, and local regulations. Stringent data protection measures, as per EU Regulation 2016/679, are in place. Informed consent is obtained from parents or legal guardians and documented throughout the study. Relevant safety information will be provided to the appropriate IECs during the study in accordance with local regulations and requirements. In the event of unexpected early termination of the study for any unanticipated reason, the investigator will be responsible for informing the IRB/IEC of the early termination

### 4.9 Publication policy

Findings will be published in peer-reviewed journals

### 5. Discussion

Unlike robust data of mepolizumab's efficacy and safety in adults, current paediatric data is limited [31,32]. Studies in children aged 6–11 were either open-label, focusing on drug safety and pharmacokinetics rather than clinical effectiveness, or lacked randomisation. Double-blind, placebo-controlled trials are difficult in paediatrics due to the disease's rarity, making recruitment challenging. Moreover, parental concerns complicate enrolment in trials. *Gupta et al 's* study had a small sample size and had no placebo arm, limiting efficacy assessment [32]. Consequently, regulatory approval for mepolizumab by the FDA and EMA was based in part on extrapolation from adult data, though both agencies deemed the benefit–risk profile acceptable for children and adolescents. Real-life clinical evidence of mepolizumab in children and adolescents with SA is virtually non-existent. In a small case series of 16 adolescents with eosinophilic asthma mepolizumab was noted to reduce asthma attacks in frequency or severity for the majority [36]. Our observational study will focus on broader paediatric population, aiming to assess the effectiveness of mepolizumab in a more diverse group, including those with co-existing conditions. By capturing outcomes in a real-world clinical setting, the study aims to provide a better understanding of the treatment's impact compared to evidence derived from controlled clinical trials.

### **CASAM group

#### Spain

- Ines de Mir Messa (Hospital Universitari Vall Hebron, Barcelona. Institut de Recerca de l'Hospital Universitari Vall d'Hebron (VHIR), Barcelona): Participated in patient recruitment, data collection and study oversight. Chief Investigatory and lead author from Spain (ines.mir@vallhebron.cat)

- José Valverde-Molina (Hospital General Universitario Santa Lucía, Cartagena. IMIB Biomedical Research Institute, Murcia): Participated in patient recruitment and data collection

- Yolanda Aliaga (Hospital Universitario Miguel Servet, Zaragoza): Participated in patient recruitment and data collection

- Alicia Callejón Callejón (Hospital Nuestra Señora de Candelaria, Santa Cruz de Tenerife): Participated in patient recruitment and data collection

- Pilar Caro Aguilera (Hospital Universitario Regional de Málaga, Málaga): Participated in patient recruitment and data collection

- Silvia Castillo Corullón (Hospital Clínico Universitario, Valencia): Participated in patient recruitment and data collection

- Paula Corcuera Elosegui (Hospital Universitario Donostia, San Sebastian): Participated in patient recruitment and data collection

- Olga de La Serna Blazquez (Hospital Universitario La Paz, Madrid): Participated in patient recruitment and data collection

- Carolina Díaz García (Hospital General Universitario Santa Lucía, Cartagena): Participated in patient recruitment and data collection

- Ana Díez Izquierdo (Hospital Universitari Vall Hebron, Barcelona. Institut de Recerca de l'Hospital Universitari Vall d'Hebron (VHIR), Barcelona): Participated in patient recruitment and data collection

- Mirella Gaboli (Hospital Virgen del Rocio, Sevilla): Participated in patient recruitment and data collection

- Alvaro Gimeno Díaz De Atauri (Hospital Universitario 12 de Octubre, Madrid): Participated in patient recruitment and data collection

- Alejandro López Neyra (Hospital Universitario Niño Jesus, Madrid): Participated in patient recruitment and data collection

- Jaime Lozano Blasco (Hospital Sant Joan de Deu, Barcelona): Participated in patient recruitment and data collection

- Ana Martínez Cañavate (Hospital Virgen de las Nieves, Granada): Participated in patient recruitment and data collection

- Lorena Moreno Requena (Hospital Virgen de las Nieves, Granada): Participated in patient recruitment and data collection

- Juan Navarro Morón (Hospital Germans Trias i Pujol, Badalona): Participated in patient recruitment and data collection

- Laura Valdesoiro Navarrete (Hospital Parc Tauli, Sabadell): Participated in patient recruitment and data collection

**United Kingdom**

- Atul Gupta (King's college hospital, King's college London): Participated in patient recruitment, data collection and study oversight. Chief Investigatory and lead author from UK (atul.gupta@kcl.ac.uk)

- Andrew Turnbull (Great Ormond Street Hospital, London): Participated in patient recruitment and data collection

- Chinedu Nwokoro (Royal London Hospital, London): Participated in patient recruitment and data collection

- Clare S Murray (Royal Manchester Children's Hospital, Manchester): Participated in patient recruitment and data collection

- Katharine Pike (Department of Respiratory Paediatrics, Bristol Royal Hospital for Children, Bristol): Participated in patient recruitment and data collection

- Dr Angela Tang (Department of Respiratory Paediatrics, Bristol Royal Hospital for Children): Participated in patient recruitment and data collection
- Prasad Nagakumar (Birmingham Women's and Children's Hospital, Birmingham): Participated in patient recruitment and data collection

## What is already known on this topic:

- Severe asthma is life threatening and has poor outcomes in terms of morbidity, mortality, and cost.
- Mepolizumab, as an add-on maintenance therapy for severe eosinophilic asthma, has demonstrated efficacy in clinical trials, reducing asthma exacerbations and enhancing asthma control in patients with this phenotype.

## What this study hopes to add

- To assess the impact and long-term adverse effects of mepolizumab in children & adolescents with severe asthma with diverse demographics, co-morbidities and concomitant asthma medications use in routine clinical practice.

## Limitations

- Several potential limitations were acknowledged, including challenges posed by the COVID-19 pandemic, enrolment bias, channelling bias, follow-up bias, and potential impact on study validity.

## Supporting information

**S1 File. S5 File spirit checklist.**
(DOCX)

## Acknowledgments

The authors would like to thank all patients/guardians who will contribute to the performance of the study and all the CASAM investigators/centres for their invaluable help in the patient recruitment and data collection

## Author contributions

**Conceptualization:** Atul Gupta, Ines de Mir Messa, Jose Valverde-Molina.

**Data curation:** Atul Gupta, Ines de Mir Messa, Jose Valverde-Molina, Clare S Murray, Luis Moral, Javier Torres Borrego, Katharine Pike, Ana Díaz Izquierdo, Ana Martínez-Cañavate, Prasad Nagakumar, James Cook, Latika Gupta.

**Funding acquisition:** Atul Gupta, Jose Valverde-Molina.

**Investigation:** Atul Gupta, Ines de Mir Messa, Jose Valverde-Molina.

**Methodology:** Atul Gupta, Ines de Mir Messa, Jose Valverde-Molina.

**Project administration:** Atul Gupta, Ines de Mir Messa, Jose Valverde-Molina, Clare S Murray, Luis Moral, Javier Torres Borrego, Katharine Pike, Ana Díaz Izquierdo, Ana Martínez-Cañavate, Prasad Nagakumar, James Cook, Latika Gupta.

**Resources:** Atul Gupta, Ines de Mir Messa, Jose Valverde-Molina.

**Supervision:** Atul Gupta, Ines de Mir Messa, Jose Valverde-Molina, Clare S Murray, Luis Moral, Javier Torres Borrego, Katharine Pike, Ana Díaz Izquierdo, Ana Martínez-Cañavate, Prasad Nagakumar, James Cook, Latika Gupta.

**Validation:** Atul Gupta, Ines de Mir Messa, Jose Valverde-Molina.

**Visualization:** Atul Gupta, Ines de Mir Messa, Jose Valverde-Molina.

**Writing – original draft:** Atul Gupta, Ines de Mir Messa, Jose Valverde-Molina.

**Writing – review & editing:** Atul Gupta, Ines de Mir Messa, Jose Valverde-Molina, Clare S Murray, Luis Moral, Javier Torres Borrego, Katharine Pike, Ana Díaz Izquierdo, Ana Martínez-Cañavate, Prasad Nagakumar, James Cook, Latika Gupta.

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
