## [Decision Letter · Decision Letter 0]

2 Sep 2025

Dear Dr. Gupta,

We look forward to receiving your revised manuscript.

Kind regards,

Stelios Loukides

Academic Editor

PLOS ONE

Journal Requirements:

“This research was supported by an Investigator-Sponsored Studies (ISS) GSK grant, with no additional contributions to the project.”

“This research was supported by an Investigator-Sponsored Studies (ISS) GSK grant, with no additional contributions to the project.”

4. One of the noted authors is a group or consortium [CASAM Group]. In addition to naming the author group, please list the individual authors and affiliations within this group in the acknowledgments section of your manuscript. Please also indicate clearly a lead author for this group along with a contact email address.

6. Please ensure that you refer to Figure 1 & 2 in your text as, if accepted, production will need this reference to link the reader to the figure.

7. We note you have included a table to which you do not refer in the text of your manuscript. Please ensure that you refer to Table 1-3 in your text; if accepted, production will need this reference to link the reader to the Table

Reviewers' comments:

Reviewer's Responses to Questions

**Comments to the Author**

1. Does the manuscript provide a valid rationale for the proposed study, with clearly identified and justified research questions?

Reviewer #1: Yes

Reviewer #2: Yes

2. Is the protocol technically sound and planned in a manner that will lead to a meaningful outcome and allow testing the stated hypotheses?

Reviewer #1: Yes

Reviewer #2: Yes

3. Is the methodology feasible and described in sufficient detail to allow the work to be replicable?

Reviewer #1: Yes

Reviewer #2: Yes

4. Have the authors described where all data underlying the findings will be made available when the study is complete?

Reviewer #1: No

Reviewer #2: Yes

5. Is the manuscript presented in an intelligible fashion and written in standard English?

Reviewer #1: Yes

Reviewer #2: Yes

You may also provide optional suggestions and comments to authors that they might find helpful in planning their study.

Reviewer #1: This protocol presents a multictre observational study that will evaluates the management of severe eosinophilic asthma in children and adolescents with mepolizumab. The introduction states the burden of paediatric severe asthma and the limited trial data on mepolizumab for children and adolescents. The primary objective is to assess the reduction in clinically significant exacerbations pre- and post-treatment. Methodologically, the study is feasible and technically sound.

While data management and long-term storage procedures are detailed, a data-sharing plan (eg repository, conditions of access, timeline) has not been specified.

In the exclusion criteria I believe that the following should be included:

1)Known hypersensitivity to mepolizumab or excipients.

2)Active clinically significant infection, eg untreated helminth/parasitic infection

3)Other primary pulmonary diseases that may influence outcomes according the investigator (eg cystic fibrosis, primary ciliary dyskinesia, interstitial lung disease)

4)Eosinophilic conditions with different indications

5)Concomitant biologic therapy for asthma or recent exposure without washout

Finally, the severity of the asthma exacerbations (mild, moderate, severe) should be a secondary objective

Reviewer #2: I have reviewed the "Protocol for an open labelled observational study in children & adolescents with severe asthma with an eosinophilic phenotype treated with mepolizumab (CASAM)" which is regarding a study initiated in March 2022 and likely to complete in March 2026. The study primarily aims to establish the efficacy of mepolizumab in eosinophilic severe asthma patients in the age group 6-17 years. It is an open label, real world observational study.

It is a very well written protocol especially designed to meet ethical issues that may arise in this population. All the aspects of methodology have been described in sufficient detail and utilize contemporary tools. Considering its scientific rigour, study will generate valuable data.

**Do you want your identity to be public for this peer review?** For information about this choice, including consent withdrawal, please see our Privacy Policy

Reviewer #1: No

Reviewer #2: No

---

## [Author Response · Author response to Decision Letter 1]

2 Oct 2025

Response to editors and reviewers:

1. A rebuttal letter: has been provided and uploaded

2. 'Revised Manuscript with Track Changes': uploaded

3. An unmarked version 'Manuscript': uploaded

4. Funding statement changed as advised in cover letter

5. Financial disclosure: changed as advised in cover letter

6. Group authorship:all members of the group are acknowledged in manuscript acknowledgement section

7. Ethics statement: is in methodology section, numbered now

8. Please ensure that you refer to Figure 1 & 2 in your text as, if accepted, production will need this reference to link the reader to the figure: manuscript updated

9. Please ensure that you refer to Table 1-3 in your text; if accepted, production will need this reference to link the reader to the Table: updated in manuscript

10. Data availability statement changed in manuscript.

---

## [Decision Letter · Decision Letter 1]

12 Oct 2025

Protocol for an open labelled observational study in children & adolescents with severe asthma with an eosinophilic phenotype treated with mepolizumab (CASAM)

PONE-D-25-35399R1

Dear Dr. Gupta,

We’re pleased to inform you that your manuscript has been judged scientifically suitable for publication and will be formally accepted for publication once it meets all outstanding technical requirements.

Kind regards,

Stelios Loukides

Academic Editor

PLOS ONE

Additional Editor Comments (optional):

Reviewers' comments:

Reviewer's Responses to Questions

**Comments to the Author**

1. Does the manuscript provide a valid rationale for the proposed study, with clearly identified and justified research questions?

Reviewer #1: Yes

2. Is the protocol technically sound and planned in a manner that will lead to a meaningful outcome and allow testing the stated hypotheses?

Reviewer #1: Yes

3. Is the methodology feasible and described in sufficient detail to allow the work to be replicable?

Reviewer #1: Yes

4. Have the authors described where all data underlying the findings will be made available when the study is complete?

Reviewer #1: Yes

5. Is the manuscript presented in an intelligible fashion and written in standard English?

Reviewer #1: Yes

You may also provide optional suggestions and comments to authors that they might find helpful in planning their study.

Reviewer #1: This revised manuscript presents a protocol for a multicenter observational study evaluating the management of severe eosinophilic asthma in children and adolescents treated with mepolizumab. We thank the authors for addressing the reviewers’ comments.

**Do you want your identity to be public for this peer review?** For information about this choice, including consent withdrawal, please see our Privacy Policy

Reviewer #1: No

---

## [Editor Report · Acceptance letter]

PONE-D-25-35399R1

PLOS ONE

Dear Dr. Gupta,

I'm pleased to inform you that your manuscript has been deemed suitable for publication in PLOS ONE. Congratulations! Your manuscript is now being handed over to our production team.

Kind regards,

on behalf of

Dr. Stelios Loukides

Academic Editor

PLOS ONE